# Distinct Akt phosphorylation states are required for insulin regulated Glut4 and Glut1-mediated glucose uptake

**Muheeb Beg[1†], Nazish Abdullah[1†], Fathima Shazna Thowfeik[2,3], Nasser K Altorki[2,3], Timothy E McGraw[1,2,3]***

[1]Department of Biochemistry, Weill Cornell Medicine, New York, United States; [2]Department of Cardiothoracic Surgery, Weill Cornell Medicine, New York, United States; [3]Lung Cancer Program, Meyer Cancer Center, Weill Cornell Medicine, New York, United States

**Abstract** Insulin, downstream of Akt activation, promotes glucose uptake into fat and muscle cells to lower postprandial blood glucose, an enforced change in cellular metabolism to maintain glucose homeostasis. This effect is mediated by the Glut4 glucose transporter. Growth factors also enhance glucose uptake to fuel an anabolic metabolism required for tissue growth and repair. This activity is predominantly mediated by the Glut1. Akt is activated by phosphorylation of its kinase and hydrophobic motif (HM) domains. We show that insulin-stimulated Glut4-mediated glucose uptake requires PDPK1 phosphorylation of the kinase domain but not mTORC2 phosphorylation of the HM domain. Nonetheless, an intact HM domain is required for Glut4-mediated glucose uptake. Whereas, Glut1-mediated glucose uptake also requires mTORC2 phosphorylation of the HM domain, demonstrating both phosphorylation-dependent and independent roles of the HM domain in regulating glucose uptake. Thus, mTORC2 links Akt to the distinct physiologic programs related to Glut4 and Glut1-mediated glucose uptake.

*For correspondence: temcgraw@med.cornell.edu

[†]These authors contributed equally to this work

**Competing interests:** The authors declare that no competing interests exist.

## Introduction

The Akt family of kinases are a major node in cell signaling, with regulatory roles in numerous physiologic processes, including cell growth, proliferation, differentiation, metabolism, cell migration and survival (*Manning and Cantley, 2007*). Perturbations in Akt signaling are linked to many diseases, from hyper-activation in cancer to blunted activity in metabolic diseases (*Franke, 2008*).

There are three Akt homologues: Akt1, Akt2 and Akt3 (also referred to as protein kinase B (PKB)α, -β and -γ, respectively). All three share a common domain structure consisting of an amino-terminal pleckstrin homology (PH) domain, a kinase domain, and a carboxyl-terminal regulatory region referred to as the hydrophobic motif (HM) domain (*Chan et al., 1999*). The canonical mode of Akt activation involves plasma membrane recruitment to sites of phosphoinositol 3-kinase (PI3K) activity by Akt PH domain binding to phosphoinositide 3,4,5 trisphosphate (PIP3), the product of growth factor activation of PI3K (*Vivanco and Sawyers, 2002*). Concurrent with plasma membrane recruitment, Akt is phosphorylated in its kinase domain at T308/9 (residue number for Akt1 and Akt2, respectively) by phosphoinositide-dependent protein kinase 1 (PDPK1), which is itself recruited to the plasma membrane by its own PH domain (*Frech et al., 1997*). Full Akt activation is associated with a second phosphorylation on S473/4 in the HM domain by mTORC2 (*Sarbassov et al., 2005*). The exact link between growth factor stimulation and activation of mTORC2 activity is not known (*Sparks and Guertin, 2010*). The kinase domain phosphorylation, T308/9, is key for activation

(*Alessi et al., 1996*), whereas S473/4 phosphorylation of the HM domain allosterically enhances kinase activity (*Alessi et al., 1997*) and contributes to substrate selectivity (*Jacinto et al., 2006*).

A major function of Akt is to transmit insulin signaling to the control of cellular metabolism. One well characterized biological output of Akt is its role in insulin-stimulated translocation of Glut4 glucose transporter to the plasma membrane of fat and muscle cells (*Jiang et al., 2003*). The redistribution of Glut4 underlies enhanced glucose uptake responsible for postprandial blood glucose lowering (*Abel et al., 2001*; *Zisman et al., 2000*). Akt also mediates growth factor stimulated glucose uptake into cells other than fat and muscle (*Ward and Thompson, 2012*). In those cases, enhanced glucose uptake is to fulfill more local needs, such as fueling an anabolic metabolism required for cell growth and tissue repair; compared to the effect of insulin on fat and muscle cells, which is to maintain whole body glucose homeostasis. Enhanced Glut1-mediated glucose uptake by cancer cells is a prominent example of growth factor-stimulated glucose uptake untethered from regulation of whole body glucose homeostasis (*Carvalho et al., 2011*).

Akt is a principal mediator of growth factor action. However, technical impediments have made it challenging to define Akt isoform-specific roles as well as the precise roles of the two activating phosphorylations in signal transduction. We developed a system to study ectopically expressed Akt without interference from endogenous Akt (*Kajno et al., 2015*). The ectopically expressed Akt is engineered to contain a mutation in the PH domain (W80A) that confers resistance to MK2206, an allosteric Akt inhibitor (*Calleja et al., 2009*; *Green et al., 2008*). Acute inhibition of endogenous Akt's with MK2206 allows for functional studies of the ectopically expressed Akt, while minimizing compensatory changes due to loss of Akt functions. We have previously used this system to study Akt isoform specificity in adipogenesis (*Kajno et al., 2015*).

Here we use this system to define the roles of Akt2 T309 and S474 phosphorylations in insulin-regulated glucose uptake by adipocytes and proliferative cells. Phosphorylation of T309 by PDPK1 is required for Glut4 translocation, whereas phosphorylation of S474 by mTORC2 is not required. However, insulin-stimulated Glut1 translocation to the plasma membrane of adipocytes and proliferative cells is dependent on both T309 and S474 phosphorylations, linking growth factor regulation of Glut1-mediated glucose uptake to mTORC2 activity.

## Results

To reveal the roles of Akt2 T309 and S474 Akt phosphorylations in insulin-stimulated Glut4 translocation, we studied Akt2-W80A, a mutant resistant to the allosteric pan Akt inhibitor MK2206. We quantitatively determined Glut4 translocation to the plasma membrane of adipocytes in studies of a HA-Glut4-GFP reporter, an established functional assay of insulin activity in fat and muscle cells (*Karylowski et al., 2004*; *Lampson et al., 2001*; *Zeigerer et al., 2002*; *Zhao et al., 2009*; *Boguslavsky et al., 2012*). Throughout this work, unless noted otherwise, we refer to cultured 3T3-L1 adipocytes simply as adipocytes. Although adipocytes express both Akt1 and Akt2, we focused on Akt2 since it has previously been shown to be the Akt isoform biased towards regulation of glucose metabolism (*Leavens et al., 2009*). Akt2-W80A and HA-Glut4-GFP plasmids were transiently co-expressed in adipocytes by electroporation (*Kajno et al., 2015*). The Akt2-W80A construct contains a Flag epitope tag at the amino terminus and co-expression of Akt2-W80A and HA-Glut4-GFP was confirmed by GFP fluorescence and anti-FLAG immunofluorescence. Greater than 90% of the transfected cells co-expressed both proteins. To determine the expression of the constructs relative to the native proteins, we assessed total Glut4 and Akt by quantitative immunofluorescence and compared that value between cells expressing and not expressing the constructs. HA-Glut4-GFP expression was determined by GFP fluorescence and Akt2-W80A by anti-Flag staining. The cells expressing HA-Glut4-GFP showed about 1.5 fold increase in total Glut4 staining (*Figure 1—figure supplement 1*), while cells expressing Akt2-W80A showed about two fold increase in total Akt (*Figure 1—figure supplement 2*). These data demonstrate that the constructs are expressed to about the same level as native proteins.

A 30 min incubation with 1 nM insulin promoted an approximate 10-fold increase of Glut4 in the plasma membrane of control adipocytes, and co-expression of Akt2-W80A did not affect insulin response (*Figure 1C*). A 60 min pre-incubation with 1 µM MK2206 abolished insulin-stimulated Glut4 translocation in control adipocytes, whereas Glut4 translocation was unaffected by MK2206 in cells expressing Akt2-W80A (*Figure 1C*).

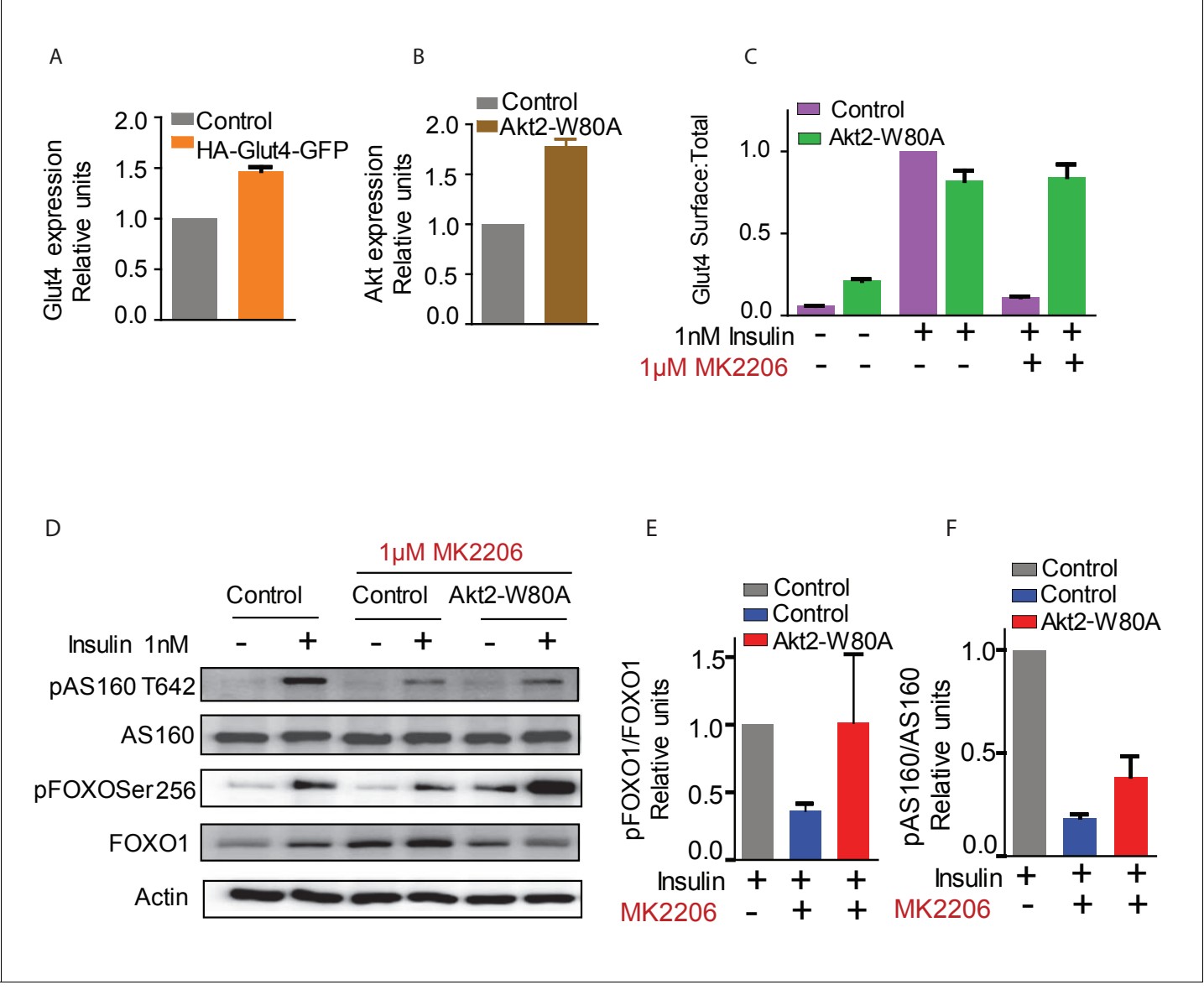

**Figure 1.** Function of ectopic Akt can be selectively studied by expression of Akt2-W80A and using Akt inhibitor MK2206. (**A and B**) Relative expression levels of HA-Glut4-GFP reporter (**A**) and Akt2-W80A (**B**) relative to endogenous proteins in 3T3-L1 adipocytes used in this study. Cultured 3T3-L1 adipocytes were electroporated with HA-Glut4-GFP or Akt2-W80A. The cells were then stained with antibodies against the endogenous protein to measure the expression of ectopic protein relative to the endogenous. GFP fluorescence (HA-Glut4-GFP) or FLAG staining (Akt2-W80A) was used to identify cells expressing the ectopic construct. Akt expression depicts total Akt expression. Data are normalized to the expression level in control adipocytes. Data shows average of two independent experiments± SEM. (**C**) Insulin stimulated translocation of HA-Glut4-GFP in control adipocytes or adipocytes expressing Akt2-W80A with or without the Akt inhibitor MK2206. Cells were stimulated with insulin for 30 min. To inhibit Akt, cells were treated for 60 min with MK2206 before insulin treatment. Data are normalized to the Glut4 surface expression in control cells treated with 1 nM insulin. Data shows average of three independent experiments± SEM. (**D**) Phosphorylation of Akt substrates AS160 and FOXO1 under the conditions used in (**C**). (**E and F**) quantification of AS160 and FOXO1 phosphorylation from (**D**).

The following figure supplements are available for figure 1:

**Figure supplement 1.** Adipocytes showing expression levels of ectopic Glut4, relative to endogenous Glut4.

**Figure supplement 2.** Adipocytes showing expression levels of ectopic Akt2-W80A, relative to endogenous Akts.

We next determined the activity of Akt2-W80A based on phosphorylation of its substrates AS160 and FoxO1. AS160 (TBC1D4), a Rab GTPase activating protein (GAP), is an Akt target required for Glut4 translocation (*Miinea et al., 2005*; *Sano et al., 2003*; *Geraghty et al., 2007*; *Eguez et al., 2005*). Expression of Akt2-W80A fully rescued Akt phosphorylation of FoxO1 and partially rescued Akt T642 phosphorylation in cells in which native Akts are inhibited by MK2206 (*Figure 1D,E,F*). The Akt2-W80A partial rescue of AS160 phosphorylation, in context of its complete rescue of Glut4 translocation (*Figure 1C*), indicates that AS160 phosphorylation is not linearly correlated with Akt regulation of Glut4 translocation.

## Akt phosphorylation of AS160 is not a reliable surrogate of physiological activity

We next performed a more detailed investigation to determine the degree to which Akt phosphorylation of AS160 is uncoupled in Akt-dependent Glut4 translocation. MK2206 in a dose-dependent manner, inhibited insulin-stimulated Glut4 translocation in control adipocytes, with an EC50 of less than 0.1 µM MK2206 (*Figure 2A*). Adipocytes co-expressing Akt2-W80A were resistant to a dose up to 1 µM MK2206. Thus, 1 µM MK2206 can be used to distinguish ectopically expressed Akt2-W80A activity from the activities of endogenous Akt's. The insulin dose-response of Glut4 translocation in MK2206-treated adipocytes expressing Akt2-W80A, was similar to that in untreated control adipocytes, validating this as an experimental system to quantitatively assess Akt function downstream of insulin stimulation (*Figure 2B*).

Although Glut4 translocation was fully blocked by 1 µM MK2206, AS160 phosphorylation on T642 was only partially inhibited (*Figure 2C and D*). Thus, Akt phosphorylation of AS160-T642 is not a linear readout of Akt control of Glut4 translocation. Insulin-stimulated Akt phosphorylation on T309 and S474, and phosphorylation of AS160 on T642 were unaffected by 1 µM MK2206 in adipocytes expressing Akt2-W80A, in agreement with the results of Glut4 translocation (*Figure 2C and D*). Based on these results, we chose 1 µM MK2206 for subsequent structure-function studies of Akt2-W80A. As was the case for the insulin-dose response measured by Glut4 translocation, MK2206 did not affect the insulin dose-response of Akt2-W80A phosphorylation on T309 and S474, confirming the system can be used to quantitatively assess Akt activation downstream of insulin-stimulation (*Figure 2E and F*).

## T309 phosphorylation, but not S474 phosphorylation, is required for insulin-stimulated Glut4 translocation and mTORC1 activation

Activation of Akt2 involves phosphorylation of T309 and S474 (*Sarbassov et al., 2005*; *Gonzalez and McGraw, 2009a*). To probe the requirements of these phosphorylations for Akt2 signaling to Glut4, we studied the effects of alanine substitutions at these sites. Total Akt in cells expressing the mutants was about twice that in untransfected cells, demonstrating the mutants, like Akt2-W80A (*Figure 1B*), were expressed at about the same level as total native Akts (*Figure 3A*). Mutation of both T309 and S474 blocked insulin-stimulated Glut4 translocation, as did the single mutation of T309, demonstrating that T309 phosphorylation is necessary for Akt2 to signal to Glut4 (*Figure 3B*). Surprisingly, Glut4 translocation was fully supported by Akt2-W80A-S474A mutant, establishing that S474 phosphorylation is not required for insulin-stimulated Glut4 translocation (*Figure 3B*). Akt2-T309A and Akt2-S474A mutants were expressed to similar levels, demonstrating that the functional difference between these mutants was not due to difference in the expression levels (*Figure 3A*).

Insulin-stimulated phosphorylation of S474 was unaffected by T309A mutation, demonstrating T309 phosphorylation is not a prerequisite for S474 phosphorylation (*Figure 3C and D*). Similarly, the S474A mutation did not affect T309 phosphorylation (*Figure 3C and D*). Thus, in adipocytes these phosphorylations are independent of one another.

Akt2-W80A-S474A supported insulin-stimulated phosphorylation of S6 kinase, an mTORC1 target downstream of Akt (*Figure 3E and F*). Thus, in addition to Glut4 translocation, Akt activation of mTORC1 does not require phosphorylation of S474. As expected, the T309A mutation blunted insulin-stimulated phosphorylation of S6 kinase, consistent with phosphorylation of T309 being required for Akt activity (*Figure 3E and F*).

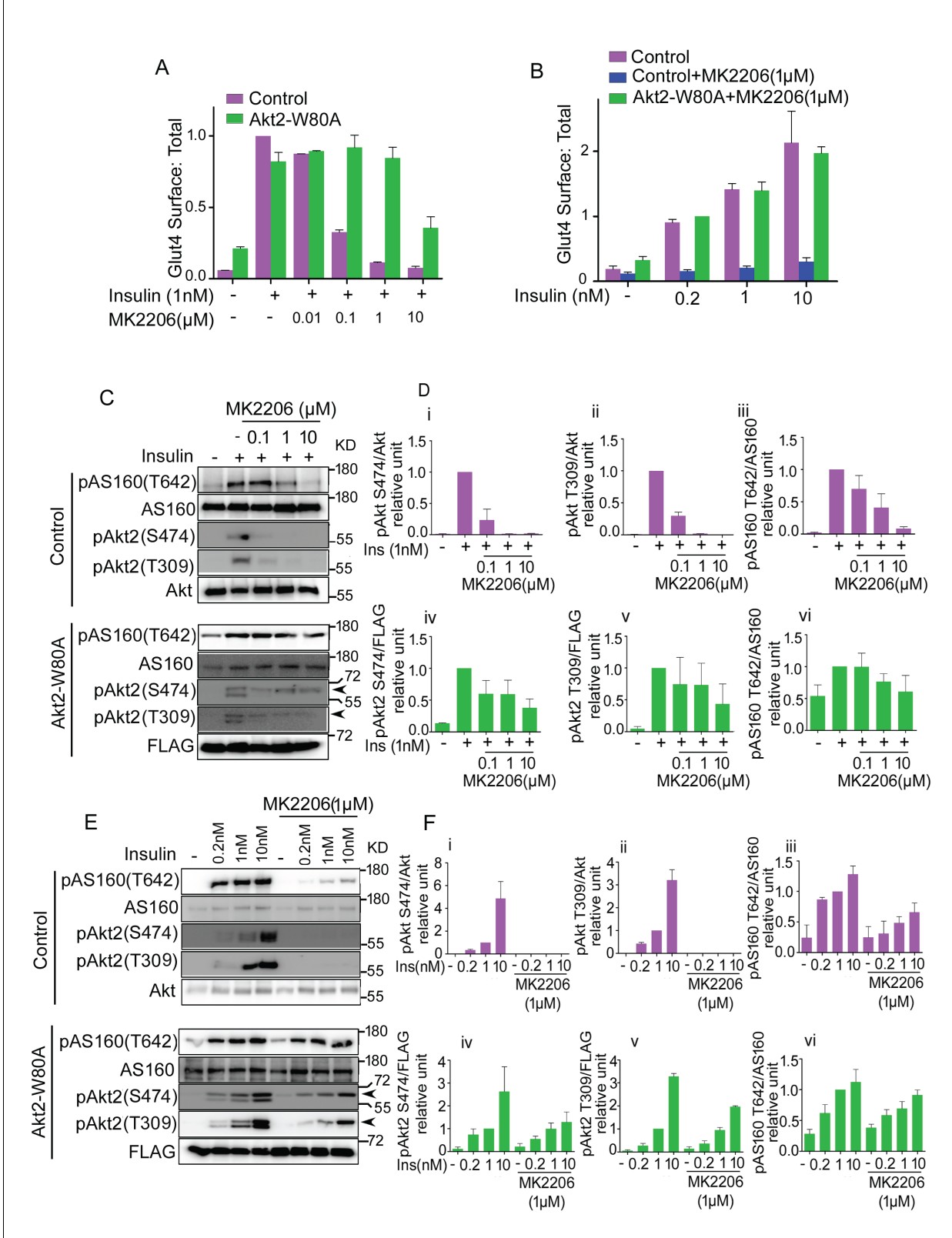

**Figure 2.** Akt substrate phosphorylation does not directly reflect its physiological activity. (**A and B**). Optimization of the dose of MK2206 and insulin to study insulin stimulated Glut4 translocation in 3T3-L1 adipocytes. Surface (Cy3 anti-HA) to total (GFP) ratio of HA-GLUT4-GFP is plotted in control adipocytes or adipocytes co-expressing Akt-W80A. Cells were pretreated with the indicated dose of MK2206 for 1 hr followed by 30 min stimulation by insulin. More than 30 cells were quantified per condition per assay. Mean normalized values ± SEM. n = 3–4 independent experiments. In panel A, the

*Figure 2 continued on next page*

*Figure 2 continued*

data of the individual experiments are normalized to Glut4 surface to total value in control cells stimulated with 1 nM insulin. In panel B, the data are normalized to Glut4 surface to total value in Akt2-W80A + MK2206, 0.2 nM insulin condition. (**C and E**). In a similar experimental setup, cell lysates were collected and subjected to immunoblot analysis for Akt and AS160 phosphorylation. Arrow heads note the migration of ectopically expressed Akt, whose migration is slower due to the amino-terminal Flag epitope. (**D and F**). The quantification for (**C**) and (**E**) respectively. (i-iii) control adipocytes, (iv-vi) adipocytes transiently transfected with ectopic Akt. Each data normalized to 1 nM insulin. n = 5 independent experiments.

Insulin-stimulated Glut4 translocation promoted by endogenous Akt (cells not treated with MK2206) was unaffected in adipocytes expressing either T309A or S474A mutants compared to control adipocytes (*Figure 3G*). These data establish that the mutants are not dominant-interfering of endogenous Akt.

## Inhibition of PDPK1 and knockdown of mTORC2 phenocopy the Akt2 phosphorylation mutants

PDPK1 is responsible for phosphorylation of T309, and mTORC2 for phosphorylation of S474 (*Sarbassov et al., 2005*; *Alessi et al., 1997*; *Jacinto et al., 2006*). The activities of the Akt2 phosphorylation mutants predict that PDPK1 but not mTORC2 is required for insulin-stimulated Glut4 translocation. Consistent with this hypothesis, transient knockdown of mTORC2 activity by targeting the Rictor subunit had no effect on Glut4 translocation, despite reducing insulin-stimulated phosphorylation of S474 by approximately 80% (*Figure 4A–D*). As expected, mTORC2 knockdown did not affect phosphorylation of T309.

The PDPK1 inhibitor, GSK2334470, blocked insulin-stimulated Glut4 translocation in a dose-dependent manner (*Figure 4E*). The inhibition of translocation correlated with reduced T309 phosphorylation (*Figure 4F*). GSK2334470 did not affect insulin-stimulated phosphorylation of S474. These results reinforce the conclusion that S474 phosphorylation is dispensable, whereas T309 phosphorylation is essential for Glut4 translocation in adipocytes.

## Constitutive activity of Akt2 S474D mutation is dependent on T309 phosphorylation

Several studies have demonstrated that substitution of aspartate for S474 (S474D) generates a constitutively active Akt, presumably as a phosphomimetic (*Yang et al., 2002a*, *2002b*; *Hart and Vogt, 2011*). In agreement with that previous work, Akt2-W80A-S474D promoted Glut4 translocation in the absence of insulin stimulation (*Figure 5A*). Consistent with the functional activity, T309 was constitutively phosphorylated in the S474D mutant; that is, without insulin stimulation (*Figure 5B*). The Akt2-W80A-S474D-induced Glut4 translocation was blocked by T309A mutation as well as by PDPK1 inhibition (*Figure 5A and D*). Thus, the aspartate substitution at S474 alters Akt2 in a way that enhances phosphorylation of T309 by basal activity of PDPK1, resulting in Akt2 activation and Glut4 translocation, yet phosphorylation of S474 is not required for Glut4 translocation.

## Constitutive activity of membrane targeted Akt2 depends on T309 phosphorylation

The recruitment of Akt to the plasma membrane is a crucial step in Akt activation. A naturally occurring mutation in the Akt PH domain, E17K, enhances association with the plasma membrane, independent of stimulated PI3 kinase activity, leading to constitutive Akt activity and Glut4 translocation in adipocytes (*Gonzalez and McGraw, 2009b*; *Carpten et al., 2007*). Increased plasma membrane Glut4 induced by expression of Akt2-E17K-W80A was blocked by the T309A mutation as well as by the PDPK1 inhibitor GSK2334470, providing independent evidence that Glut4 translocation is dependent on PDPK1 phosphorylation of T309 (*Figure 5F*). The S474A mutation had no effect on the increased plasma membrane Glut4 induced by expression of Akt2-E17K-W80A, consistent with Glut4 translocation being independent of S474 phosphorylation (*Figure 5E*).

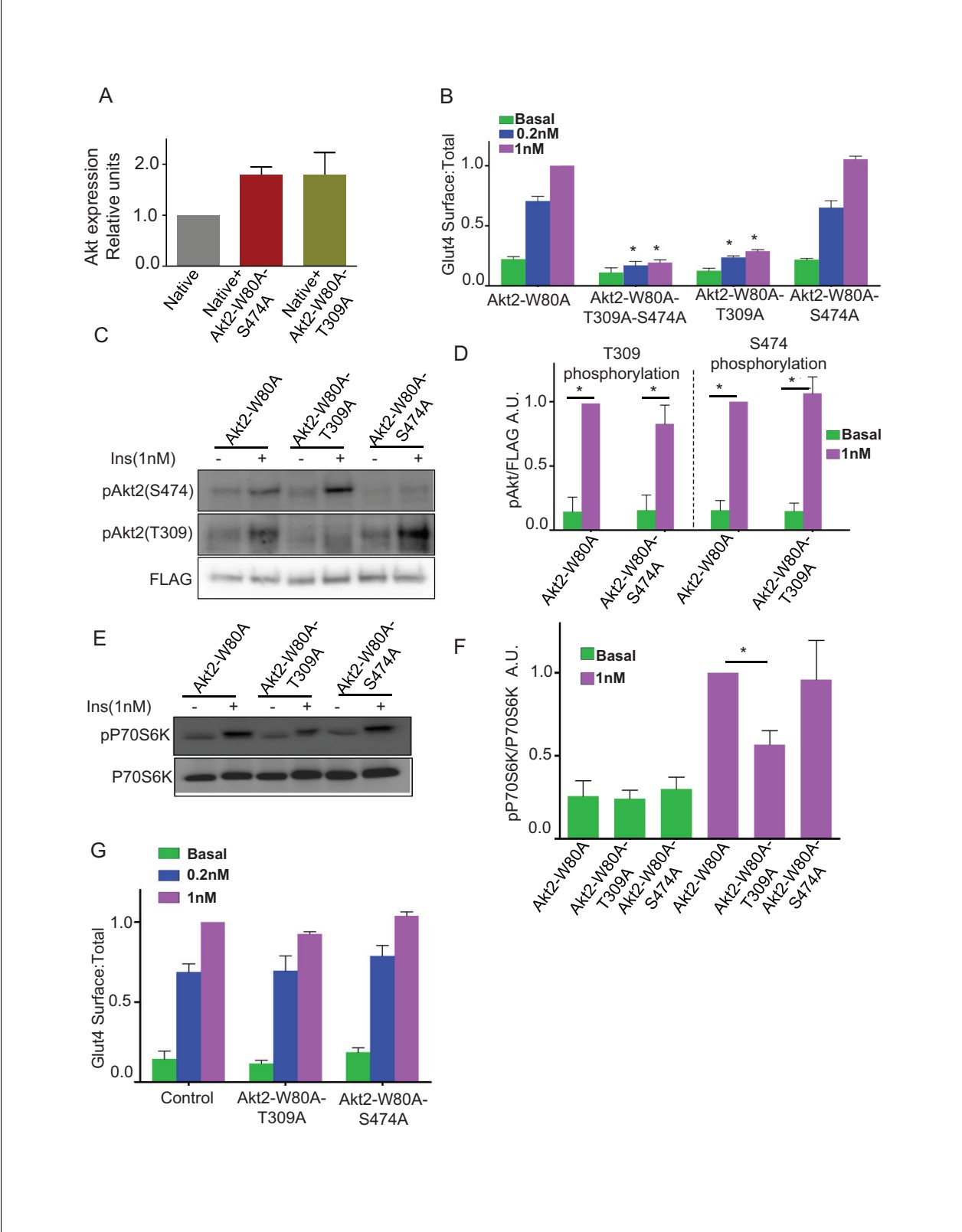

**Figure 3.** Phosphorylation of T309 is required and sufficient for insulin effect on GLUT4 and mTORC1. (**A**) Relative levels of expression of Akt2-W80A-S474A or Akt2-W80A-T309A in cells electroporated with the respective construct. Expression levels are measured with antibody that identify total Akt similar to *Figure 1A*. Data are normalized to the expression level in control adipocytes. Data shows average of 2 independent experiments ± SEM. (**B**) Quantification of surface-to-total ratio of HA-Glut4-GFP in 3T3-L1 adipocytes transiently co-expressing indicated mutants of Akt2. Cells were pretreated

*Figure 3 continued on next page*

*Figure 3 continued*

with MK2206 for 1 hr followed by 30 min insulin stimulation. More than 30 cells quantified per condition per assay. Mean normalized values ± SEM. n = 5–7 independent experiments. The data of the individual experiments are normalized to Glut4 surface to total value in control cells stimulated with 1 nM insulin. *p<0.05 compared to the respective Glut4 surface to total value in Akt2-W80A cells. (C) Representative immunoblot for Akt phosphorylation on T309 or S474 in cells expressing indicated Akt2 mutants. FLAG expression is shown as the level of expression of the Akt2 mutants. (D) Quantitation of blots similar to panel B. Averages ± SEM are plotted normalized to insulin stimulated condition. n = 6 independent experiments. *p<0.05. (E) P70-S6 kinase phosphorylation was assessed in cells expressing indicated Akt mutants. (F) Quantitation of blots similar to panel D. Average ± SEM are plotted normalized to insulin stimulated Akt2-W80A. n = 3 independent experiments. *p<0.05. (G) Quantification of surface-to-total HA-Glut4-GFP in control adipocytes or adipocytes co-expressing Akt2-W80A-T309A or Akt2-W80A-S474A. Cells without insulin stimulation (basal) or stimulated for 30 min with insulin were studied without inhibition of native Akt's by MK2206. Data are means ± SEM, n = 2.

## Complementation of HM domain deletion by membrane targeting of the HM Domain in trans

S474 phosphorylation has been linked to target selection (*Jacinto et al., 2006*). To determine if the HM domain, which contains S474, is required for Akt2 to support Glut4 translocation, we studied the activities of panel of Akt2 domain deletion constructs (*Figure 6A*). Akt2 in which the HM domain was deleted (Akt2-kinase domain construct) did not support insulin-stimulated Glut4 translocation, despite the deletion construct being expressed to the same level as full length Akt2-W80A (*Figure 6B,C*). T309 in Akt2-kinase domain construct was phosphorylated in insulin-stimulated cells, demonstrating the HM domain is not required for PDPK1 to phosphorylate Akt2 T309 but the HM domain is required for Glut4 translocation (*Figure 6C*). In addition, Akt in which the PH domain was deleted (kinase-HM domains) did not support Glut4 translocation, confirming that plasma membrane targeting is required (*Figure 6C*).

The HM domain provided in trans did not rescue Glut4 translocation in cells expressing Akt2-kinase domain construct (*Figure 6B*). However, co-expression of the HM domain with a PH domain fused to its amino terminus (PH-HM domain), rescued insulin-stimulated Glut4 translocation activity to the Akt2-kinase domain construct (*Figure 6B*). Expression of the PH-HM domain alone had no effect on Glut4 translocation. Thus, the enhanced co-localization of the Akt2 kinase domain and the HM domain achieved when both are independently targeted to plasma membrane sites of PI3 kinase activity was required for complementation of the HM domain in trans. Despite this functional complementation, S474 of the PH-HM domain construct was not phosphorylated in insulin-stimulated adipocytes, providing further support that S474 phosphorylation is not required for Glut4 translocation.

## Phospho-S474 Akt2 promotes glucose uptake by increasing GLUT1 in the plasma membrane

To confirm that phosphorylation of S474 is not required for translocation of native Glut4, we generated adipocytes stably expressing Akt2-W80A or Akt2-W80A-S474A. We established that both stably expressed Akt2-W80A and Akt2-W80A-S474A support translocation of HA-Glut4-GFP reporter. To monitor the behavior of native Glut4, we measured insulin-stimulated glucose uptake, the biological output of Glut4 translocation to the plasma membrane. Unexpectedly, insulin-stimulated glucose uptake in Akt2-W80A-S474A adipocytes was only 70% of that in cells expressing Akt2-W80A, despite Akt2-W80A-S474A supporting full translocation of Glut4 to the plasma membrane (*Figure 7A and B*). The difference in glucose uptake was not due to changes in expression of endogenous Glut4 between the Akt2-W80A or Akt2-W80A-S474A adipocytes (*Figure 7C*).

Adipocytes also express Glut1, a widely-expressed glucose transporter whose function is not specifically linked to insulin control of blood glucose. Glut1 is used by most cells for 'housing keeping' glucose uptake. Nonetheless, insulin, as well as other growth factors, promote an increase of Glut1 in the plasma membrane of many cell types (*Lawrence et al., 1992*; *Egert et al., 1999*; *Rett et al., 1996*; *Clarke et al., 1994*). To explore the behavior of Glut1 in adipocytes, we quantified plasma membrane expression using an HA-tagged Glut1 construct, a reporter used previously in studies of Glut1 trafficking (*Chaudhary et al., 2016*; *Takenouchi et al., 2007*; *Lee et al., 2015*). In cells expressing HA-Glut1, total Glut1 was about four times that in untransfected cells, demonstrating HA-Glut1 was expressed to about three times the level of native Glut1 (*Figure 7—figure*

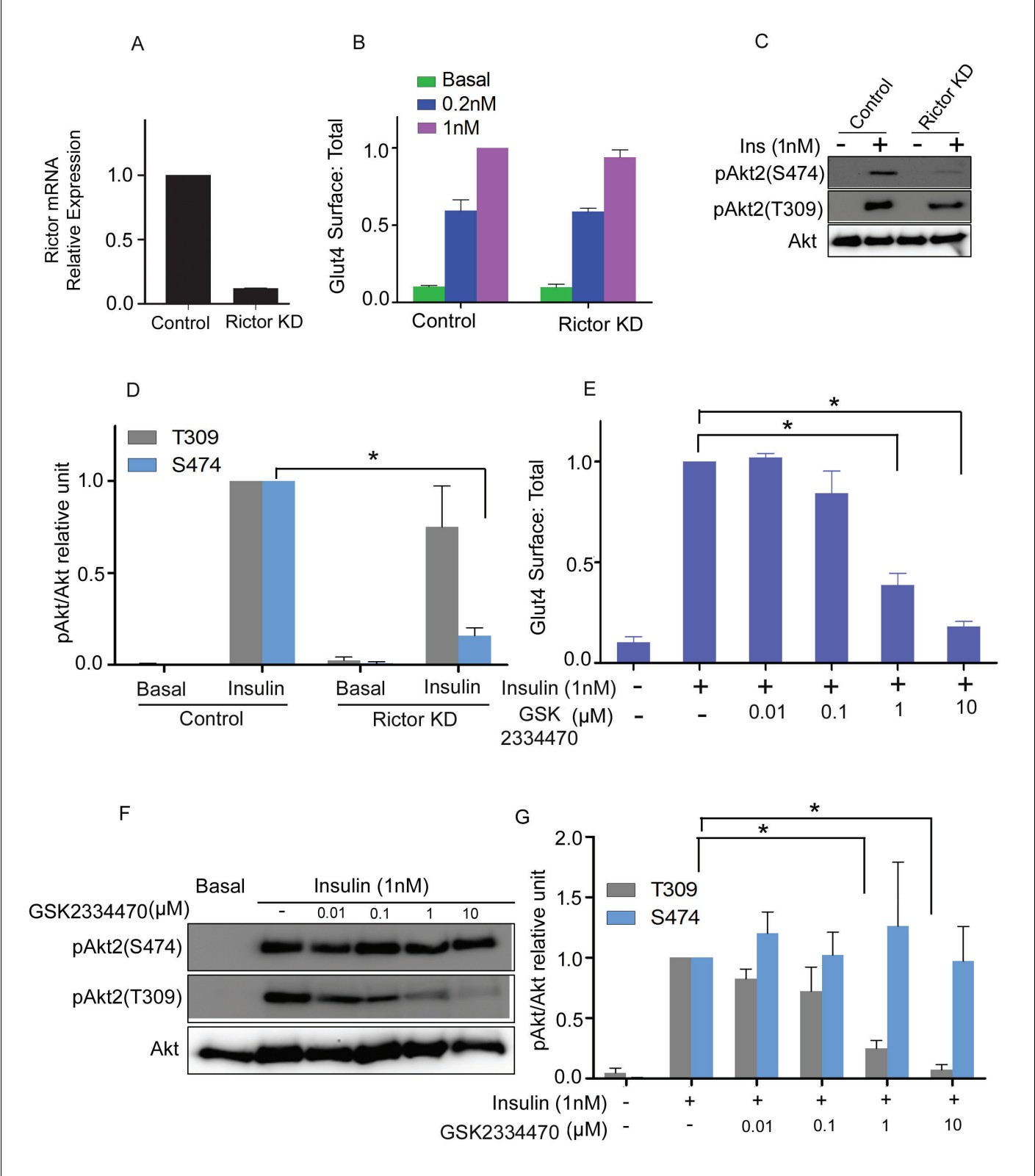

**Figure 4.** mTORC2 is not required for insulin's regulation of Glut4. (**A**) 3T3-L1 adipocytes were transfected with Rictor siRNA and cell lysates collected 48 hr later. RT-PCR was performed for Rictor knockdown assessment. n = 3 independent experiments. (**B**) Quantification of surface-to-total ratio of HA-Glut4-GFP in control adipocytes and rictor knockdown adipocytes in basal and insulin treated conditions. Mean normalized values ± SEM. n = 3 independent experiments. The data of the individual experiments are normalized to Glut4 surface to total value in control cells stimulated with 1 nM

*Figure 4 continued on next page*

*Figure 4 continued*

insulin. (C) Immunoblot to assess Akt phosphorylation (S474/T309) in control and Rictor knockdown cells. Representative blot. (D) Quantitation of blots similar to (C). Mean ± SEM normalized to 1 nM insulin treated control cells is plotted. n = 4 independent assays. *p<0.05. (E) Quantification of surface-to-total ratio of HA-Glut4-GFP in control adipocytes and adipocytes pretreated for 30 min with indicated doses of PDPK1 inhibitor GSK2334470, prior to insulin stimulation for next 30 min. Mean normalized values ± SEM. n = 5 independent assays. The data of the individual experiments are normalized to Glut4 surface to total value in control cells stimulated with 1 nM insulin. *p<0.05. (F) Representative immunoblots assessing insulin-stimulated phosphorylation of Akt after 30 min pretreatment with the indicated doses of GSK2334470. (G) Quantitation of blots similar to (F). Average is plotted normalized to 1 nM insulin treated control cells. n = 4 independent assays. *p<0.05.

*supplement 1*). Akt2-W80A supported a near 2-fold translocation of HA-Glut1 to the plasma membrane, whereas Akt2-W80A-S474A did not support insulin-induced Glut1 translocation. Thus, unlike Glut4, S474 phosphorylation was required for insulin to signal to Glut1 (*Figure 7E*). Akt2-W80A-T309A did not support Glut1 translocation, consistent with phosphorylation of T309 being required for activity (*Figure 7E*).

To link glucose uptake to Glut1 expression in the plasma membrane, we used Indinavir, an HIV protease inhibitor with an off-target effect of directly inhibiting Glut4-mediated glucose uptake (*Murata et al., 2002*; *Rudich et al., 2003*). Indinavir blunted insulin-stimulated glucose uptake downstream of both Akt2-W80 and Akt2-W80A-S474A, consistent with the bulk of increased glucose uptake being mediated by Glut4 (*Figure 7F*). Importantly, the differential in glucose uptake between adipocytes expressing Akt2-W80A and Akt2-W80A-S474A was unaffected by Indinavir, consistent with the reduced insulin-stimulated glucose uptake supported by S474A is due to reduced Glut1-mediated glucose transport. There were no differences in expression of endogenous Glut1 between adipocytes expressing Akt2-W80A or Akt2-W80A-S474A, confirming reduced glucose uptake was due a defect in translocation rather than reduced Glut1 expression (*Figure 7—figure supplement 2*).

## Phospho-S474-Akt2-dependent increased plasma membrane Glut1 is independent of AS160 inactivation and is not due to effects on general endocytic recycling

AS160 is the principal Akt target required for insulin-stimulated Glut4 translocation (*Eguez et al., 2005*; *Mîinea et al., 2005*; *Sano et al., 2003*). Akt phosphorylation inhibits AS160 GAP function, resulting in Rab10-dependent Glut4 translocation (*Sano et al., 2007*). An AS160 mutant in which 4 of its Akt phosphorylation sites are mutated to alanine (AS160-4A) is a dominant inhibitor of insulin-stimulated Glut4 translocation (*Zeigerer et al., 2004*). AS160-4A did not inhibit Glut1 translocation (*Figure 8A*). These data demonstrate Akt2 promotes increased plasma membrane Glut1 independent of its effect on Glut4, consistent with previous studies demonstrating Glut1 is not trafficked to the plasma membrane through the same pathway used by Glut4 (*Lee et al., 2015*; *Palmada et al., 2006*; *Olsen et al., 2014*).

Insulin promotes an increase in general membrane recycling back to the plasma membrane, revealed by increased transferrin receptor on the plasma membrane (*Subtil et al., 2000*; *Johnson et al., 1998*). This effect was not dependent on S474 phosphorylation since Akt2-W80-S474A supports insulin effects on the transferrin receptor (*Figure 8B*). Thus, insulin-stimulated Glut1 translocation is not an effect of Akt2 on the general endocytic recycling pathway, demonstrating that a pathway specifically downstream of Akt2 phospho-S474 targets Glut1 trafficking.

## Akt2 S474 phosphorylation is required for Glut1 translocation in proliferative cells

Glucose uptake in proliferative cells is predominantly mediated by Glut1, and enhanced plasma membrane expression of Glut1 contributes to increased glucose metabolism fueling the anabolic metabolism of cancer cells (*Chan et al., 2011*). To determine if Akt2-S474 phosphorylation has a role in plasma membrane expression of Glut1 in cancer cells, we transiently co-expressed HA-Glut1 and Akt2-W80A or Akt2-W80A-S474A in a genetically engineered mouse lung cancer cell line (KP1 cells) (*Choi et al., 2015*). Insulin stimulated a near 2-fold increase of Glut1 in the plasma membrane of KP1 cells co-expressing Akt2-W80A but not in cells co-expressing Akt2-W80A-S474A (*Figure 8C*).

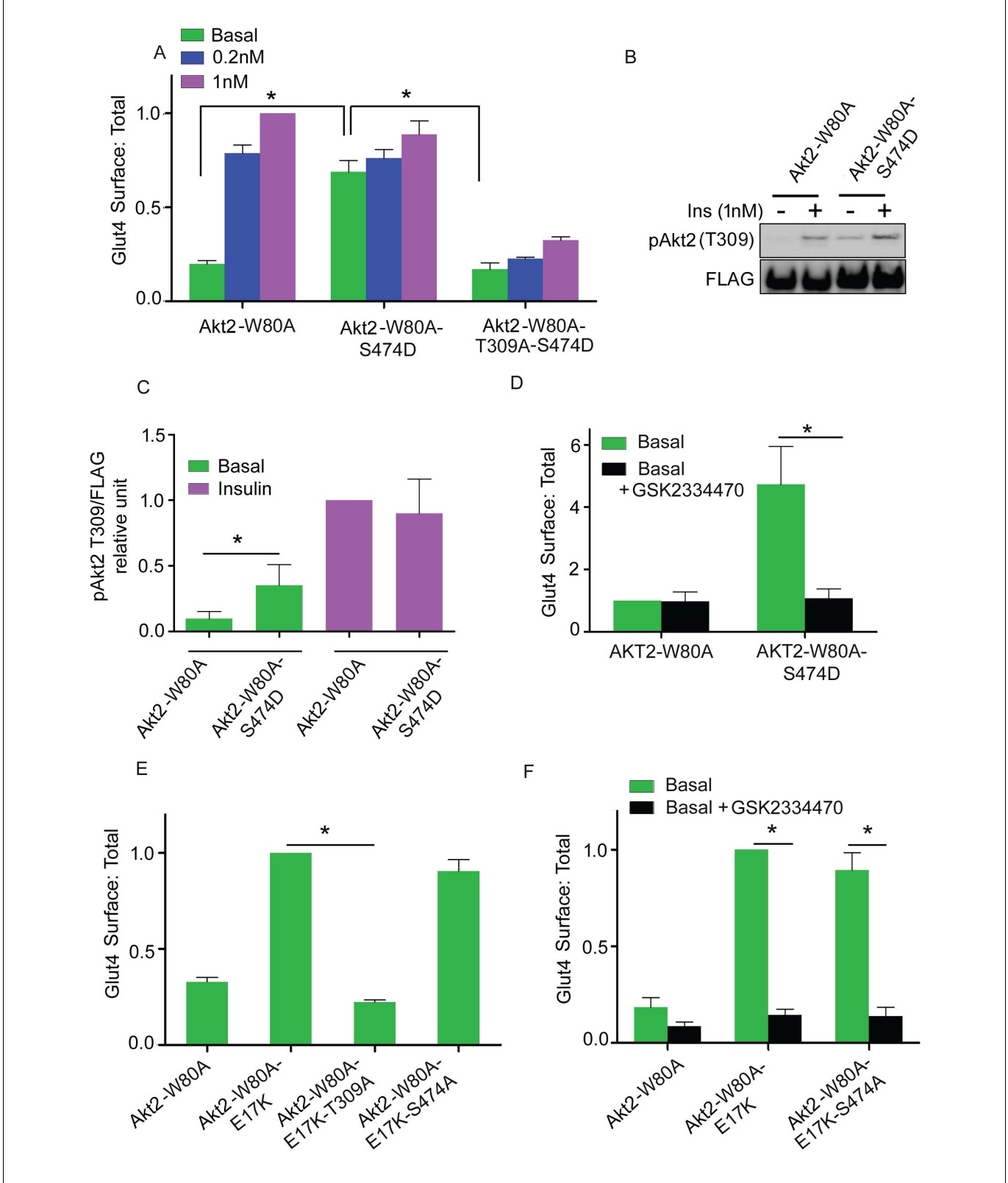

**Figure 5.** Akt2 constitutively active mutants require T309 phosphorylation. (**A**) Quantification of surface-to-total ratio of HA-Glut4-GFP in 3T3-L1 adipocytes transiently co-expressing indicated Akt2 mutants. Cells were pretreated with MK2206 for 1 hr followed by 30 min insulin stimulation. More than 30 cells were quantified per condition per assay. Mean normalized values ± SEM. n = 3–4 independent assays. The data of the individual experiments are normalized to Glut4 surface to total value in control cells stimulated with 1 nM insulin. *p<0.05. (**B**) Representative immunoblot of T309

*Figure 5 continued on next page*

*Figure 5 continued*

phosphorylation in cells expressing Akt2-W80A or Akt2-W80A-S474D. Cells were pretreated with MK2206 followed by 1 nM insulin for 30 min. Cell lysates were subjected to western blotting to assess phosphorylation. FLAG expression was used as measure of Akt2 expression. (C) Quantitation of blots similar to (B). Mean ± SEM is plotted normalized to insulin stimulated Akt2-W80A. n = 3 independent assays. *p<0.05. (D) Quantification of surface-to-total ratio of HA-Glut4-GFP in cell co-expressing Akt2-W80A or Akt2-W80A-S474D under basal state. Cells were pretreated with MK2206 for 1 hr where GSK2334470, a PDPK inhibitor co-incubated for last 30 min. More than 30 cells quantified per condition per assay. Mean normalized values ± SEM. n = 3 assays. The data of the individual experiments are normalized to Glut4 surface to total value in basal control cells stimulated. *p<0.05. (E) Quantification of surface-to-total ratio of HA-Glut4-GFP in cells co-expressing Akt2-W80A or constitutively active mutant Akt2-W80-E17K and its T309A and S474A mutants. Cells were pretreated with MK2206 for 1 hr and Glut4 Surface to total determined in basal condition. More than 30 cells quantified per condition per assay. Mean normalized values ± SEM. n = 4 independent assays. The data of the individual experiments are normalized to Glut4 surface to total value in cells expressing Akt2-W80A-E17K. *p<0.05. (F) Quantification of surface-to-total ratio of HA-Glut4-GFP in cells co-expressing indicated mutants under basal state. Cells were pretreated with MK2206 for 1 hr where GSK2334470, a PDPK1 inhibitor co-incubated for last 30 min. More than 30 cells quantified per condition per assay. n = 3 assays. The data of the individual experiments are normalized to Glut4 surface to total value in cells expressing Akt2-W80A-E17K. *p<0.05.

Thus, insulin-stimulated Glut1 translocation is downstream of Akt S474 phosphorylation in this cancer cell line.

To investigate the role of Akt2-S474 phosphorylation in Glut1-mediated glucose uptake, we generated KP1 cells stably expressing Akt2-W80A or Akt2-W80A-S474A. KP1 cells express constitutively active KRAS-G12D, which drives constitutive activity of Akt downstream of PI3 kinase. In KP1 cells stably expressing Akt2-W80A or Akt2-W80A-S474A, MK2206 treatment inhibits endogenous Akt, and therefore, any Akt-mediated effects will be downstream of the ectopically expressed Akt, allowing us to query the role of S474 phosphorylation in KRAS driven Glut1 plasma membrane expression. Glucose uptake by Akt2-W80A-S474A expressing KP1 cells was reduced compared to Akt2-W80A expressing KP1 cells, consistent with a role for phosphorylation of S474 in controlling the amount of Glut1 in the plasma membrane (*Figure 8D*). These data demonstrate a role for S474 phosphorylation in Glut1 expression on the plasma membrane downstream of KRAS, revealing a role for mTORC2-Akt in the regulation of Glut1-depedent glucose metabolism in proliferative cells.

## Discussion

The Akt kinases are a central node in cell signaling. The two activating phosphorylations of Akt provide a mechanism by which Akt activity and signaling integrates the outputs of PDPK1 and mTORC2. The roles of the T308/9 and S473/4 for Akt activation and substrate selections have been analyzed in detail in in-vitro studies (*Yang et al., 2002a*; *Balzano et al., 2015*). Here we study their roles in intact adipocytes downstream of insulin, a natural stimulus. In adipocytes, one of the key biological output of insulin activation of Akt is the translocation of Glut4 to the plasma membrane, resulting in enhanced glucose uptake. PDPK1 phosphorylation of Akt2-T309 is required for insulin-stimulated Glut4 translocation since either T309A mutation or inhibition of PDPK1 inhibits translocation (*Figure 3 and 4*). These findings are in line with previous reports that T309 phosphorylation is essential for Akt activation (*Alessi et al., 1997*; *Stokoe et al., 1997*). However, Akt2 phosphorylation at S474 is not required for Glut4 translocation in adipocytes, despite the fact that this phosphorylation is robustly stimulated by insulin and therefore is often used as a sentinel of insulin-sensitivity. Neither mutation of S474 to alanine nor inactivation of mTORC2 by silencing of Rictor, affected insulin-stimulated Glut4 translocation (*Figure 3 and 4*). Thus, insulin-regulated glucose uptake by adipocytes, a critical feature for whole body glucose homeostasis, is independent of mTORC2 phosphorylation of Akt. S6 kinase phosphorylation, an Akt-dependent activity downstream of mTORC1, was also independent of S474 phosphorylation.

Past studies have shown that mTORC2 phosphorylation of S473/4 has a role in Akt substrate selection. For example, ablation of Rictor-mTORC2 depressed FoxO1/3 phosphorylation, whereas phosphorylation of Akt substrates TSC2 and GSK3 were not affected (*Jacinto et al., 2006*). Our work contributes to understanding the role of phospho-S474 Akt in substrate selection by demonstrating this phosphorylation is required for functional regulation of Glut1 but not Glut4.

The direct role of Akt S474 phosphorylation in Glut4 translocation has not been investigated in the context of native insulin regulation of Akt activity (*Chen et al., 2003*; *Ducluzeau et al., 2002*).

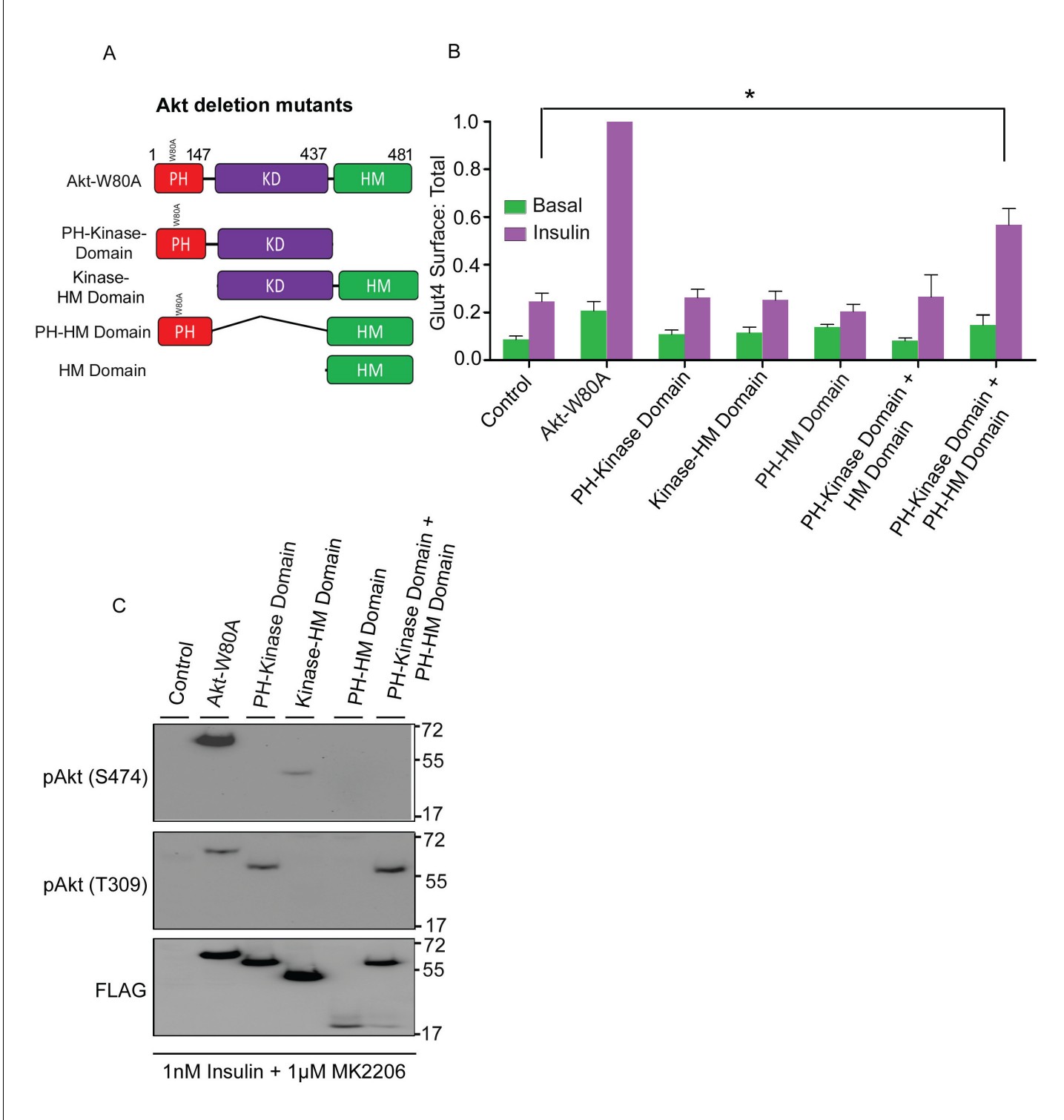

**Figure 6.** HM domain targeted to membrane can complement kinase domain in-trans to restore Akt2 activity. (**A**) Schematic representation of full-length Akt2-W80A and indicated domain deletion constructs. Numbers represent domains boundaries deduced from literature. (**B**) Quantification of surface-to-total ratio of HA-Glut4-GFP in control adipocytes or adipocytes transiently co-expressing either full length Akt2-W80A or indicated deletion mutants. Cells were pretreated with MK2206 for 1 hr followed by 30 min insulin stimulation. More than 30 cells quantified per condition per assay. Mean normalized values ± SEM. n = 6 independent assays. The data of the individual experiments are normalized to Glut4 surface to total value in Akt2-W80A expressing cells stimulated with 1 nM insulin. *p<0.05. (**C**) Representative immunoblot of Akt2 phosphorylation (T309/S474) in cells expressing full length and indicated deletion mutants. Cells were pretreated with MK2206 and stimulated with insulin for 30 min. n = 3 independent experiments.

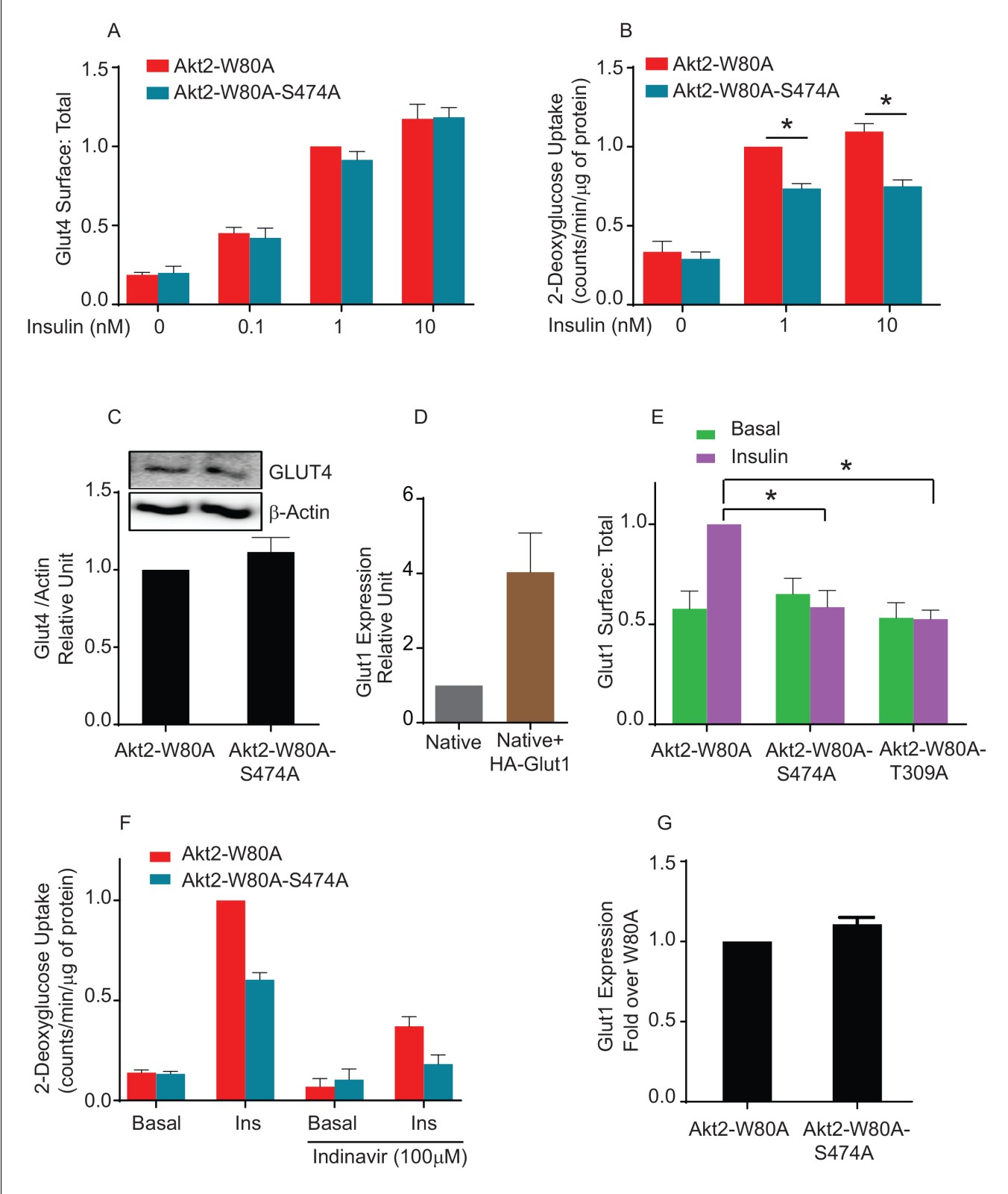

**Figure 7.** Akt2 phosphorylation on S474 is required for insulin mediated Glut1 translocation and Glut1-mediated glucose uptake. (**A**) Quantification of surface-to-total ratio of HA-Glut4-GFP in adipocyte stably expressing Akt2-W80A and Akt2-W80A-S474A. Cells were pretreated with MK2206 for 1 hr followed by indicated dose of insulin. Mean normalized values ± SEM is plotted. n = 3 independent experiments. The data of the individual experiments are normalized to Glut4 surface to total value in Akt2-W80A cells stimulated with insulin. (**B**) 2-Deoxyglucose uptake was measured in cells

*Figure 7 continued on next page*

*Figure 7 continued*

stably expressing Akt2-W80A or Akt2-W80A-S474A. Cell were pretreated with MK2206 for 1 hr followed by indicated dose of insulin for 30 min. Glucose uptake was performed in last 5 min of post insulin stimulation. $^3$H-2-deoxyglcusoe uptake was normalized to total protein content for each well in every assay. Each data further normalized to 1 nM insulin treated Akt2-W80A cells. n = 5 independent experiments. *p<0.05. (C) Representative immunoblot of Glut4 and respective actin expression in cells stably expressing W80A or S474A Akt2 mutants (inset). Bars represent quantitation of the immunoblot. Mean normalized values ± SEM is plotted. n = 3 independent experiments. (D) Relative levels of expression of HA-Glut1 in electroporated cells. Expression levels are measured with antibody that identify both endogenous and ectopic Glut1. Data are normalized to the expression level in control adipocytes. Data shows average of 2 independent experiments± SEM. (E) Quantification of surface-to-total ratio of HA-Glut1 expression in 3T3-L1 adipocytes transiently expressing indicated Akt2 mutants under basal and insulin stimulated condition. Cells were pretreated with MK2206 for 1 hr followed by insulin (170 nM) treatment. Data is normalized to insulin treated Akt2-W80A. Mean normalized values ± SEM is plotted. n = 6 independent experiments.*p<0.05. (F) Fraction of Glut4 mediated 2-Deoxyglucose uptake in cells stably expressing Akt2-W80A or Akt2-W80A-S474A was determined by incubation of cells with HIV protease inhibitor, Indinavir for 2 hr. MK2206 was co-incubated in last 1 hr during the course of indinavir incubation followed by 170 nM insulin stimulation. Glucose uptake was performed in last 5 min of post insulin stimulation. $^3$H-2-deoxyglcusoe uptake was normalized to total protein content for each well in every assay. Each data further normalized to insulin treated Akt2-W80A cells. n = 3. *p<0.05. (G) Quantitation of the expression level of endogenous Glut1 in cells stably expressing Akt2-W80A or Akt2-W80A-S474A. More than 200 cells were counted in each assay. Mean normalized values ± SEM is plotted. n = 3 independent experiments.

The following figure supplements are available for figure 7:

**Figure supplement 1.** Adipocytes showing expression levels of ectopic Glut1, relative to endogenous Glut1.

**Figure supplement 2.** Adipocytes stably expressing Akt2-W80A or Akt2-W80A-S474A showing expression levels of endogenous Glut1.

Two groups have shown a loss of Akt S473 phosphorylation and decreased insulin-stimulated adipocyte glucose uptake in primary adipocytes from adipocyte-specific Rictor knockout mice (*Tang et al., 2016*; *Kumar et al., 2010*). However, neither study established that defective insulin-stimulated Glut4 translocation was responsible for reduced glucose uptake nor did the studies agree on a mechanism for the decreased glucose uptake. Thus, our finding that S474 phosphorylation is not required for coupling of Akt activation to Glut4 translocation is compatible with the data reported for the Rictor knockouts. Furthermore, in light of our results, it is likely that the defect in glucose uptake in the Rictor knockout mice is not due to a defect in Glut4 translocation but rather due to another effect of the Rictor knockout on glucose metabolism. mTORC2 also phosphorylates several other kinases, including protein kinase A, protein kinase G and protein kinase C (*Laplante and Sabatini, 2012*).

Unexpectedly, we found that Akt2 S474 phosphorylation was required for insulin-stimulated Glut1 translocation. Both translocation of Glut1 to the plasma membrane of adipocytes and its contribution to increased glucose uptake were S474 phosphorylation-dependent (*Figure 7*). Insulin and other growth factors stimulate an approximate 2-fold increase of Glut1 in the plasma membranes of a variety of cell types, although how this is achieved is not known. The principal Akt substrate involved in Glut4 translocation, AS160, is not required for Glut1 translocation. The dominant-inhibitory AS160 mutant, AS160-4A, which inhibits Glut4 translocation, does not affect insulin-stimulated Glut1 translocation. The S474 phosphorylation-dependent translocation of Glut1 to the plasma membrane is not a result of Akt regulation of general endocytic recycling because insulin-regulation of transferrin receptor trafficking was not dependent on S474 phosphorylation. These data establish that Akt2 specifically regulates the amount of Glut1 in the plasma membrane by a mechanism requiring S474 phosphorylation.

The control of Glut1 plasma membrane expression is not responsible for insulin regulation of glucose homeostasis. That effect of insulin is dependent on regulation of Glut4 in adipocytes and muscle, cell types that express very little Glut1 (*Mitsumoto et al., 1991*). Insulin stimulation of glucose uptake into those cells serves the postprandial needs of whole body metabolism rather than the intrinsic needs of the fat and muscle cells. In other cell types the effect of growth factors on Glut1 expression in the plasma membrane (and the resultant stimulated glucose uptake) fulfil cell-intrinsic needs, such as fueling an anabolic metabolism to support cell growth and tissue expansion (*Olson et al., 1996*). Our findings reveal a specific role for phospho-S474 Akt in cellular glucose uptake mediated by Glut1, demonstrating that mTORC2 contributes to the regulation of glucose

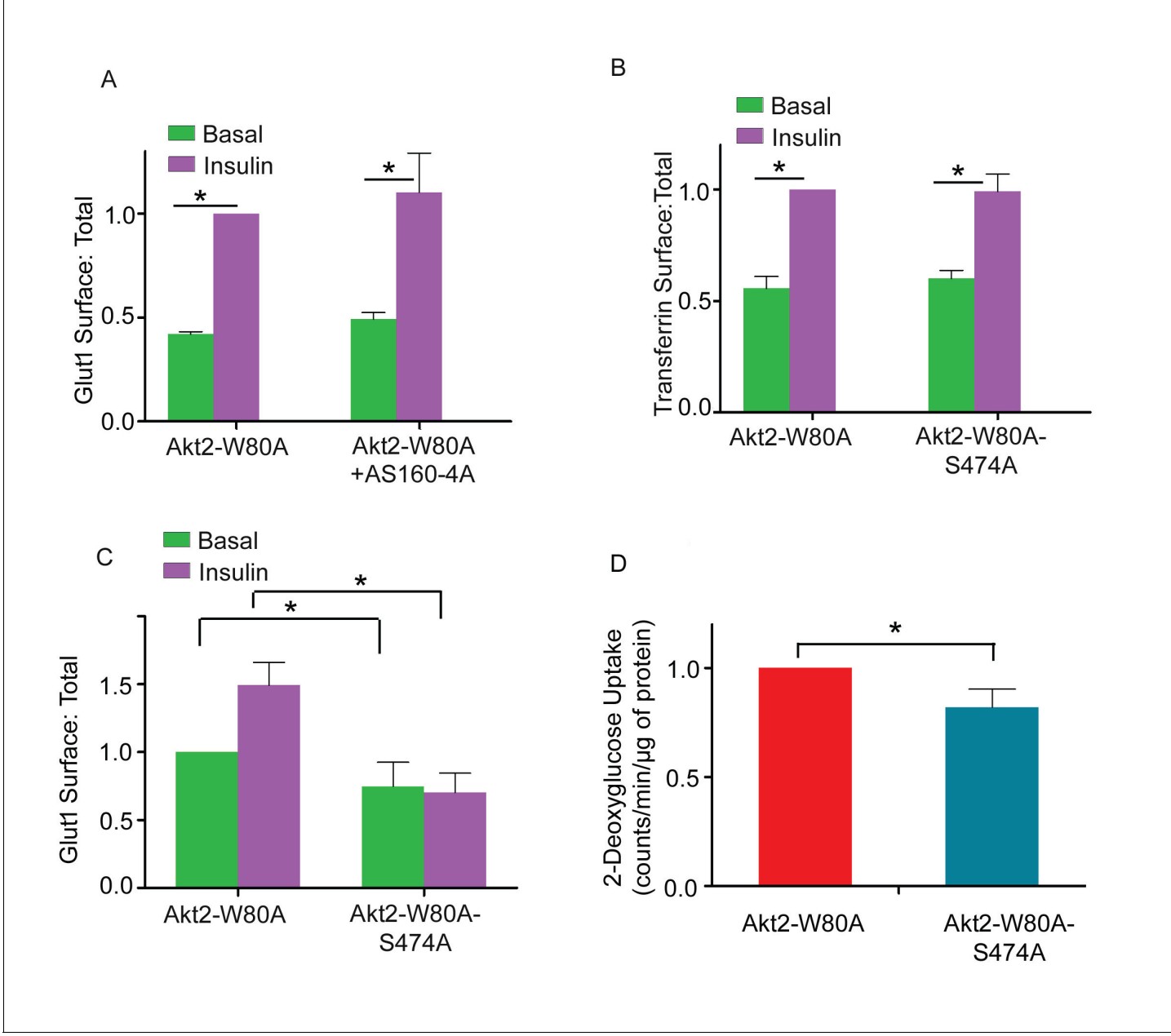

**Figure 8.** Akt2 phosphorylation on S474 mediates Glut1 trafficking and glucose uptake downstream of KRAS mutant. (**A**) Glut1 translocation was measured in 3T3-L1 adipocytes transiently expressing Akt2-W80A or co-expressing Akt2-W80A and AS160-4A. Serum-starved cells were treated with 1 μM MK2206 for one hours prior to a 30 min incubation ±170 nM insulin. HA-Glut1 reporter was used to quantify glut1 on the surface normalized to total glut1 expression. Data are mean ± SEM of 4 experiments. Data from individual experiments are normalized to the value for insulin-stimulated Akt2-W80A expressing cells. *p<0.05. (**B**) Transferrin receptor translocation was measured in response to insulin in 3T3-L1 adipocytes transiently co-expressing Akt2-W80A or Akt2-W80A-S474A and the human transferrin receptor. Serum-starved cells were treated with 1 μM MK2206 for one hours prior to a 30 min incubation ±170 nM insulin. Data are mean ± SEM of 3 experiments. Data from individual experiments are normalized to the value for insulin-stimulated Akt2-W80A expressing cells. *p<0.05. (**C**) Glut1 translocation in KP1 lung cancer cells. HA-Glut1 reporter translocation was measured in KP1 cells transiently expressing Akt2-W80A or Akt2-W80A-S474A. Serum-starved cells were treated with 1 μM MK2206 for one hours prior to a 30 min incubation ±170 nM insulin. HA-Glut1 reporter was used to quantify Glut1 on the surface normalized to total Glut1 expression. Each data has been normalized to the basal Glut1 surface:total in Akt2-W80A. n = 4 independent experiments. *p<0.05. (**D**) Basal glucose uptake in KP1 cells. KP1 cells stably expressing Akt2-W80A or Akt2-W80A-S474A were serum starved for 5 hr, with 1 μM MK2206 added for the last hour. [3]H-2-deoxyglcusoe uptake was measured during the last 2 min. Each data has been normalized to the glucose uptake in Akt2-W80A. n = 4 independent experiments. *p<0.05.

transport into cells other than those required to maintain whole body glucose homeostasis, and it does so by signaling dependent on Akt phosphorylated at S474.

Glut1 has a role in the enhanced glucose uptake characteristic of many tumor cells. This increased Glut1 activity is achieved by increased expression as well changes in Glut1 trafficking that promote accumulation of Glut1 in the plasma membrane (*Egert et al., 1999*). Akt is hyper-activated in many cancers, and our data suggest that one consequence is enhanced expression of Glut1 in the plasma membrane based on phospho-S474 Akt control of Glut1 trafficking. There is an emerging appreciation of mTORC2's role in cancer, specifically in the control of glycolytic metabolism (*Masui et al., 2014*). mTORC2, via phospho-S474 Akt, may promote Glut1-mediated glucose uptake and therefore contribute to enhanced glycolytic metabolism.

SIN1, a component of mTORC2 complex, is phosphorylated by PDPK1-activated Akt, which results in activation of mTORC2, and activated mTORC2 then phosphorylates Akt (*Yang et al., 2015*). This positive feedforward loop was demonstrated in a number of cell contexts and it may have a role to induce full Akt activation. However, this feedforward regulation is not required for all Akt regulated pathways since as we have shown here, S474A mutation did not quantitatively affect insulin-stimulated Glut4 translocation or mTORC1 activation (i.e., S6 kinase phosphorylation).

Although S474 phosphorylation, which is within the HM domain, is not required to promote Glut4 translocation, the HM domain itself is required for Akt activity. Deletion of the HM domain blocked Akt2 signaling to Glut4 translocation, despite insulin-stimulated phosphorylation of T309 and providing the HM domain in trans did not complement the defect (*Figure 6*). However, co-expression of the HM domain fused to a PH domain restored insulin-stimulated Glut4 translocation. The enforced co-localization, and subsequent local concentrations of the KD and HM domains, achieved by independently targeting the kinase and HM domains to plasma membrane sites of PI3 kinase activity is sufficient to reconstitute Akt function in Glut4 translocation. Although function is restored, S474 is not phosphorylated. These data reinforce the importance of the HM domain for Akt action and are consistent with the modular nature of Akt activation and substrate selection (*Yang et al., 2002a*).

The loss of function T309A mutation and the gain of function S474D mutation provide additional evidence that insulin activation of Akt is necessary and sufficient to induce Glut4 redistribution to the plasma membrane of adipocytes. The activity of S474D without insulin stimulation demonstrates that Akt can be activated based on the unstimulated levels of PI3 kinase and PDPK1 activities. Those findings raise the question of whether the activated levels of PI3 kinase and PDPK1 in growth factor-stimulated cells determine how quickly Akt is activated rather than PI3 kinase/PDPK1 stimulation determining the level of Akt activation achieved.

## Materials and methods

### Materials

Antibodies against Akt, phospho-Akt (Ser 473/4 and Thr 308/9- RRIDs- AB_329825 and AB_2255933), phospho-AS160 (Thr642), were obtained from Cell Signaling Technologies. Anti-Flag epitope antibody was purchased from Sigma-Aldrich. Anti-AS160 antibody was purchased from Millipore and anti-actin antibody was purchased from Cytoskeleton Inc. Anti-HA epitope antibody was purchased from Covance (RRID:AB_2565006). Cy3 labeled secondary antibody was obtained from Jackson ImmunoResearch Laboratories. Akt inhibitor MK-2206 was purchased from Cayman.

### Cell lines, cloning and generation of Akt2 mutants

3T3-L1 fibroblasts (RRID:CVCL_0A20) were maintained in culture and differentiated into adipocytes as previously described (*Zeigerer et al., 2002*). KP1 lung cancer cells were culture in maintained in DMEM supplemented with 10% FBS media. HA-Glut4-GFP and HA-Glut1 plasmids have been described previously (*Lampson et al., 2001*). FLAG-Akt2-W80A was subcloned into pLVX-IRES-tdTomato as described previously (*Kajno et al., 2015*) Stable cell lines for Akt2-W80A and Akt2-S474A were generated by lentivirus infection. The cDNAs were cloned into pLVX-IRES-tdTomato, viruses generated in HEK cells and 3T3-L1 fibroblasts (30–40% confluent) infected with virus particles for 48 hr. Post-infection stably expressing Akt construct cells were flow sorted on the basis of lentiviral-encoded Td-Tomato expression. The cDNA construct Flag-Akt2-W80A has been previously described (*Kajno et al., 2015*). T309A, S474A, S474D, S474D-T309A, S474A, E17K, E17K-T309A

and E17K-S474A were generated in backbone of Akt2-W80A construct. Site-directed mutagenesis was performed using QuickChange II (Agilent Technology) site directed mutagenesis kit. Akt2 domain deletion constructs were created using domain boundaries from published literature (*Balzano et al., 2015*).

3T3-L1 cells were originally a gift from Giulia Baldini, University of Arkansas, in 1993. The cells used in this study were derived from frozen stocks of those cells. The cells robustly differentiate into adipocytes, establishing they are pre-adipocytes. The cells were transcriptionally profiled 3 years ago, establishing they are mouse cells and have characteristics of primary adipocytes (unpublished). The cells have not recently been tested for mycoplasma.

The KP1 cells used were a gift from Brendon Stiles, Weill Cornell Medicine. These cells are transformed, based on in vivo tumorigenesis, and are mouse epithelial cells based on qPCR and immunofluorescence studies. The original KP1 cells received were mycoplasma positive.

The cell lines used are not among the commonly misidentified cell lines (International Cell Line Authentication Committee database).

## Electroporation of HA-Glut4-GFP, HA-Glut1 and FLAG-Akt2 mutants and Translocation Assays

Differentiated 3T3 L1 adipocyte were electroporated with 45 µg of Glut4 plasmid along with 30 µg of Akt2 construct for translocation assays (*Kajno et al., 2015*). For Glut4 translocation, cells were washed and incubated in serum-free media for 2 hr followed by incubation with pan Akt inhibitor MK2206 for 1 hr. Cells were then stimulated with indicated concentration of insulin for 30 min to achieve steady-state surface Glut4 levels. Cells were fixed with 3.7% formaldehyde for 7 min for staining and imaging (*Karylowski et al., 2004*; *Lampson et al., 2001*; *Zeigerer et al., 2002*; *Gonzalez and McGraw, 2009b*).

## Microscopy and image quantification

To restrict detection to HA-Glut4-GFP on the cell surface, fixed cells were stained with anti-HA antibody without permeabilization and HA staining was visualized with Cy3 labelled secondary antibody. Total HA-Glut4-GFP was visualized by GFP fluorescence. The ratio of the Cy3 to GFP fluorescence is the plasma membrane to total distribution of HA-Glut4-GFP. Plasma membrane to total expression of Glut1 was determined by measuring cell surface HA-Glut1 in fixed non-permeabilized cells, and total HA-Glut1 in a separate dish of identically treated cells by anti-HA staining of fixed permeabilized cells.

All epifluorescence images were collected on an inverted microscope at room temperature using a 20x air objective (Leica Microsystems, Jena, Germany) and a cooled charge-coupled device 12-bit camera. Exposure times and image quantification were performed using MetaMorph image processing software (Universal Imaging, West Chester PA), as previously described (*Lampson et al., 2001*). GFP and Cy3 fluorescence signals were background corrected and the surface (Cy3): total (GFP) HA-Glut4-GFP was calculated for each cell. At least 50 cells were counted at random. Statistical significance was calculated in each case and outliers were removed. To compare results across multiple experimental repeats, the plasma membrane to total values were normalized within each assay to the average value of the indicated condition. Two-tailed paired Student's t tests were performed on raw (non normalized) values from multiple assays.

To quantify the relative expression of ectopic Glut4, Glut1 or Akt2, cells were electroporated with respective construct and plated on coverslips. Expression was measured by antibody that recognize both the endogenous as well as ectopic construct. Cells expressing ectopic HA-Glut4-GFP construct were identified by GFP expression. Cells expressing ectopic HA-Glut1 were identified by HA staining. While, cells expressing ectopic Akt2 constructs were identified by Flag staining. Comparison was made between cells expressing only the endogenous protein with cells expressing ectopic construct as well as endogenous protein.

## Western blot analysis

3T3-L1 adipocytes were starved in serum-free DMEM with 20 mM of sodium bicarbonate, 20 mM HEPES (pH7.2) at 37°C in 5% $CO_2$ for 2 hr prior to all experiments, followed by 1 hr incubation with Akt inhibitor MK2206 (1 uM). After 30 min insulin (1 nM unless indicated otherwise) treatment in the

same media containing inhibitor, adipocytes were washed with cold media containing 150 mM NaCl, 20 mM HEPES, 1 mM CaCl2, 5 mM KCl, 1 mM MgCl2 (pH 7.2) and lysed in lysis buffer (Cell Signaling Technology). Western blot experiments were performed using standard protocols. Densitometric analysis of immunoblots was performed by using Image J software.

## 2-Deoxy-D-glucose (2-DOG) uptake assay

2-Deoxy-D-Glucose uptake assay was performed in differentiated adipocytes as described previously (*Vazirani et al., 2016*).

## RNA extraction and RT-PCR

Total RNA was extracted using RNasey mini kit (QIAGEN). Total 1 ug RNA was used to synthesize cDNA using RNA to cDNA EcoDry kit (Clontech). Relative expression determined by standard ΔΔCt method.

## Acknowledgements

This work was supported by NIH DK52852 (TEM), DK096925 (TEM), Weill Cornell Medicine Dean's Research Fund (NKA, TEM) and the Meyers Cancer Center Grant (NKA, TEM). We thank Jeremy Dittman, Eva Gonzalez, Lew Cantley, and members of the McGraw and Altorki lab for helpful comments.

## Additional information

### Funding

| Funder | Grant reference number | Author |
|---|---|---|
| National Institute for Health Research | DK52852 | Timothy E McGraw |
| National Institute for Health Research | DK096952 | Timothy E McGraw |

The funders had no role in study design, data collection and interpretation, or the decision to submit the work for publication.

### Author contributions

MB, Conceptualization, Data curation, Formal analysis, Writing—original draft; NA, Conceptualization, Formal analysis, Investigation; FST, Investigation, Writing—review and editing; NKA, Conceptualization, Writing—review and editing; TEM, Conceptualization, Resources, Formal analysis, Supervision, Funding acquisition, Writing—original draft, Project administration, Writing—review and editing

### Author ORCIDs

Timothy E McGraw, http://orcid.org/0000-0001-9748-263X

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
