## [Decision Letter]

Thank you for submitting your article "mTORC2 Phosphorylation of Akt is Required for Insulin Regulated Glut1 but not Glut4-Mediated Glucose Uptake" for consideration by *eLife*. Your article has been reviewed by two peer reviewers, and the evaluation has been overseen by a Reviewing Editor and Jonathan Cooper as the Senior Editor. The following individuals involved in review of your submission have agreed to reveal their identity: Amira Klip (Reviewer #1).

The reviewers have discussed the reviews with one another and the Reviewing Editor has drafted this decision to help you prepare a revised submission.

Summary:

Your findings that PDK1 mediated phosphorylation of Akt2-T309, but not TORC2 mediated phosphorylation of Akt2-S473, is required for Glut4 translocation to the plasma membrane and for glucose transport stimulation by insulin are of substantial interest. Additionally, the data supporting the concept that Glut1 regulation is mediated by Akt2 but does not require an AS160 dependent pathway is strong, although all these conclusions are based on expressed tagged constructs of Akt2 and Glucose transporters rather than the endogenous proteins. Your findings are consistent with previously published information showing that Richter KO does not inhibit Glut4 translocation (Diabetes 2010;59:1397-1406), such that the novelty of your data is somewhat diminished. Mechanistic data on the nature of the Glut1 vesicles and the substrate(s) of Akt required for Glut1 translocation would add to the novelty of the paper.

Essential revisions:

After careful consideration, your paper may be acceptable for publication if you can obtain additional data that significantly extends the work. Specifically, you will need to verify that the results you have obtained are physiologically relevant to the endogenous proteins you are studying by direct measurements of these endogenous proteins (Glut4, Glut1 and Akt2). The expression levels of the introduced protein constructs, Akts and Gluts, relative to their endogenous counterparts have the possibility to confound data interpretation. Thus it is unclear what the relative levels of expressed Akt and HA tagged Gluts versus that of endogenous Glut1 and Glut4 are, and this needs to be shown.

Additionally, the data of Figure 1 on Akt would benefit by a comparison of transfected and untransfected kinase states in the same gel. In this latter regard, the editors agree with the reviewers that you will need to define the actual kinase activities of Akt2 under the conditions of your experiments rather than just the phosphorylation states of the protein kinase.

---

## [Author Response]

Your findings that PDK1 mediated phosphorylation of Akt2-T309, but not TORC2 mediated phosphorylation of Akt2-S473, is required for Glut4 translocation to the plasma membrane and for glucose transport stimulation by insulin are of substantial interest. Additionally, the data supporting the concept that Glut1 regulation is mediated by Akt2 but does not require an AS160 dependent pathway is strong, although all these conclusions are based on expressed tagged constructs of Akt2 and Glucose transporters rather than the endogenous proteins. Your findings are consistent with previously published information showing that Richter KO does not inhibit Glut4 translocation (Diabetes 2010;59:1397-1406), such that the novelty of your data is somewhat diminished.

We thank the reviewers for their overall positive comments on our study.

In paragraph two of the Discussion, we discuss the two different mouse adipose-specific Rictor knockout models in the context of our results. We do not feel the novelty of our findings are lessened by those previous studies but rather that our detailed cellular analyses advance the field by providing a cellular context for interpretation of the knockouts as well as pointing a way for additional experimentation. In addition, our findings on the role Akt S474 phosphorylation for regulation of Glut1 in the plasma membrane are completely novel.

Mechanistic data on the nature of the Glut1 vesicles and the substrate(s) of Akt required for Glut1 translocation would add to the novelty of the paper.

We believe that our unexpected findings regarding mTORC2 regulation of Glut1 but not Glut4 trafficking provide a significant conceptual advance and a new direction for future investigations into the role of Akt in regulation of Glut1. Although we agree with the reviewer’s views regarding additional insight the investigation of Glut1 vesicles will provide, we feel those studies are better served in a dedicated effort. The mechanism by which Rictor-Akt regulates Glut1 trafficking does not change the main conclusions of our manuscript.

*Essential revisions:*

*After careful consideration, your paper may be acceptable for publication if you can obtain additional data that significantly extends the work. Specifically, you will need to verify that the results you have obtained are physiologically relevant to the endogenous proteins you are studying by direct measurements of these endogenous proteins (Glut4, Glut1 and Akt2).*

We have confirmed, within the experimental restraints imposed by the biology, that the phenomenon we describe for ectopically expressed HA-Glut4-GFP, HA-Glut1 and Akt are pertinent to the behaviors of the native proteins.

It is a challenge to study the translocation of native Glut4 and Glut1 into the plasma membrane because antibodies that recognize extracellular domains are not available. A significant advance in the field of Glut biology came with the development of tagged versions of the Glut’s, like HA-Glut4-GFP and HA-Glut1 to quantitatively assess the amount of Glut4 or Glut1 in the plasma membrane. Since insulin regulates glucose flux into cells by regulating the amount of Glut4 and Glut1 in the plasma membrane, rather than controlling transporter activity, these reporters measure physiologically relevant regulation.

The Glut4 and Glut1 reporters used in this manuscript have been previously described. HA-Glut4-GFP, over the past decade, has been used in numerous publications from a number of labs, in addition to mine [e.g., (Boguslavsky et al., 2012, Zhao et al., 2009)]. We have over 15 years of experience with the HA-Glut4-GFP reporter. We have published extensive characterizations and validations of the reporter (Karylowski et al., 2004, Lampson et al., 2001, Zeigerer et al., 2002). The models for Glut4 trafficking, based on studies of the HA-Glut4-GFP reporter, are supported by glucose transport studies (physiological data) of native Glut4. We have modified the manuscript by providing additional references for readers unfamiliar with past studies using the HA-Glut4-GFP reporter (Results section).

Although the HA-Glut1 has not been as extensively studied as HA-Glut4-GFP, it is frequently used to study the distribution of Glut1. Those previous studies validate HA-Glut1 to study Glut1 trafficking (Takenouchi et al., 2007, Lee et al., 2015). We have modified the manuscript by providing additional references for readers unfamiliar with past studies using the HA-Glut1 reporter (Results, subsection “Phospho-S474 Akt2 promotes glucose uptake by increasing GLUT1 in the plasma membrane”).

That said, we agree with the reviewers that it is important to extend the results to native Glut1 and Glut4. To do so we have studied native Glut1 and Glut4 by assessing glucose transport. The amount of Glut4 and Glut1 in the plasma membrane is rate limiting for cellular glucose uptake; consequently, changes in glucose uptake reflect changes in the amount of transporter in the plasma membrane. In cultured adipocytes, we have studied: 1) endogenous Glut4 and Glut1 (Figure 7), 2) we used indinavir to isolate Glut1-mediated glucose uptake from Glut4-mediated uptake (Figure 7), and 3) glucose uptake in KP1 cells to monitor plasma membrane Glut1 (Figure 7). In each case the change in glucose uptake correlates with the change in Glut behavior determined in studies of the reporters. We believe glucose transport studies address behaviors of the native Glut’s.

We used pharmacologic inhibition and siRNA knockdown to study the phosphorylation of native Akt at residues T308/9 and Ser473/4, and insulin-stimulated Glut4 translocation to correlate the effects of reduced phosphorylation of native Akt at these sites to altered Akt function (Figure 4). Rictor depletion, which abrogated insulin-stimulated phosphorylation of native Akt at Ser473/4, did not inhibit glucose translocation, whereas pharmacologic inhibition of PDPK1, inhibited native Akt Thr308/9 phosphorylation and Glut4 translocation. These data are in agreement with the behaviors of ectopically expressed Akt2 mutants, and we believe demonstrate physiologic relevance of our studies for native Akt regulation.

The expression levels of the introduced protein constructs, Akts and Gluts, relative to their endogenous counterparts have the possibility to confound data interpretation. Thus it is unclear what the relative levels of expressed Akt and HA tagged Gluts versus that of endogenous Glut1 and Glut4 are, and this needs to be shown.

We agree with the comment and we have added new data reporting the expression levels of the introduced constructs relative to the native proteins. We have measured the relative expression levels of HA-Glut4-GFP, HA-Glut1 or various Akt2 constructs that are expressed in cultured adipocytes by electroporation, mimicking the conditions used for functional studies. Expression levels were measured by quantitative fluorescence microscopy with antibodies that recognize both the ectopically expressed and endogenous proteins. In experiments to quantify HA-Glut4-GFP expression relative to endogenous Glut4, cells that express HA-Glut4-GFP were identified by GFP fluorescence. In experiments to quantify HA-Glut1 expression relative to endogenous Glut1, cells that express HA-Glut1 were identified by anti-HA staining. In experiments to quantify ectopic Akt 2, which is Flag-tagged, cells that express the introduced Akt2 were identified by anti-Flag staining. Comparison of the anti-Glut4, anti-Glut1 and anti-Akt fluorescence power in cells that do not, to those that do express the reporters provides a measure of the relative over-expression of the transfected constructs.

These data demonstrate that HA-Glut4-GFP is expressed at about the same level as native Glut4 (Figure 1). Similarly, different Akt2 constructs are also expressed at about the same level as total native Akts (Figure 1 and Figure 3). HA-Glut1 is expressed at about 3 times time the level of endogenous Glut1 (Figure 7). In addition to providing quantification in the main figures, fields of cells representative of those used to quantify expression are provided as figure supplements 1 and 2. The agreement between the behaviors of introduced Glut4 and Glut1 reporters with glucose uptake (native proteins) indicates that the level of expression of the reporters are not confounding data interpretation.

*Additionally, the data of Figure 1 on Akt would benefit by a comparison of transfected and untransfected kinase states in the same gel. In this latter regard, the editors agree with the reviewers that you will need to define the actual kinase activities of Akt2 under the conditions of your experiments rather than just the phosphorylation states of the protein kinase.*

The data requested is included in the revised manuscript. We have repeated the experiments to quantify the data from Figure 1 in the same gel (Figure 1). To define the kinase activities of Akt2, rather than simply relying on phosphorylation of Akt2, we have used substrate phosphorylation to demonstrate the enzymatic activity of the Akt2 constructs (Figure 1) The biochemical activity of the kinase as measured by western blotting for phosphorylation of two of the most relevant substrates, AS160 and FoxO1. The data demonstrate that Akt2-W80A construct rescues phosphorylation of AS160 and FoxO1 in MK2206-treated cells. These data support our results regarding the physiological activity of Akt2 (Glut4 translocation). These further support our previous conclusion that Akt phosphorylation of substrates are not necessarily linear readouts of Akt physiological functions.